# Rethinking the Reranker: Boundary-Aware Evidence Selection for Robust Retrieval-Augmented Generation

**Jiashuo Sun** [1]  **Pengcheng Jiang** [1]  **Saizhuo Wang** [2]  **Jiajun Fan** [1]  **Heng Wang** [1]  **Siru Ouyang** [1]  **Ming Zhong** [1]
**Yizhu Jiao** [1]  **Chengsong Huang** [3]  **Xueqiang Xu** [1]  **Pengrui Han** [1]  **Peiran Li** [4]  **Jiaxin Huang** [3]  **Ge Liu** [1]  **Heng Ji** [1]
**Jiawei Han** [1]

## Abstract

Retrieval-Augmented Generation (RAG) systems remain brittle under realistic retrieval noise, even when the required evidence appears in the top-$K$ results. A key reason is that retrievers and rerankers optimize solely for relevance, often selecting either trivial, answer-revealing passages or evidence that lacks the critical information required to answer the question, without considering whether the evidence is suitable for the generator. We propose `BAR-RAG`, which reframes the reranker as a boundary-aware evidence selector that targets the generator's Goldilocks Zone—evidence that is neither trivially easy nor fundamentally unanswerable for the generator, but is challenging yet sufficient for inference and thus provides the strongest learning signal. `BAR-RAG` trains the selector with reinforcement learning using generator feedback, and adopts a two-stage pipeline that fine-tunes the generator under the induced evidence distribution to mitigate the distribution mismatch between training and inference. Experiments on knowledge-intensive question answering benchmarks show that `BAR-RAG` consistently improves end-to-end performance under noisy retrieval, achieving an average gain of 10.3% over strong RAG and reranking baselines while substantially improving robustness. Our code and model are publicly available at https://github.com/GasolSun36/BAR-RAG.

*Figure 1.* Comparison between standard relevance-based rerankers and our boundary-aware evidence selection. Standard rerankers maximize relevance scores but overlook the generator's weaknesses, often encouraging shortcut learning and brittle reasoning by prioritizing trivial or answer-revealing evidence. In contrast, our method selects challenging yet solvable evidence based on generator uncertainty, promoting robust reasoning and reducing the mismatch between the evidence distributions encountered during training and inference under noisy retrieval.

## 1. Introduction

Retrieval-Augmented Generation (RAG) has achieved remarkable success on knowledge-intensive tasks by grounding large language model (LLM) outputs in retrieved evidence (Li et al., 2025b; Jiang et al., 2025). Yet RAG systems remain surprisingly brittle when retrieval results are noisy, partially relevant, or requires multi-step integration, even when the necessary facts exist somewhere in the top-$K$ results (Hsia et al., 2025; Yu et al., 2024; Cuconasu et al., 2024). In such realistic settings, LLMs often fail to synthesize scattered information and instead hallucinate plausible

[1]University of Illinois Urbana-Champaign [2]Hong Kong University of Science and Technology [3]Washington University in St. Louis [4]Texas A&M University. Correspondence to: Jiashuo Sun <jiashuo5@illinois.edu>.

*Proceedings of the 43rd International Conference on Machine Learning*, Seoul, South Korea. PMLR 306, 2026. Copyright 2026 by the author(s).

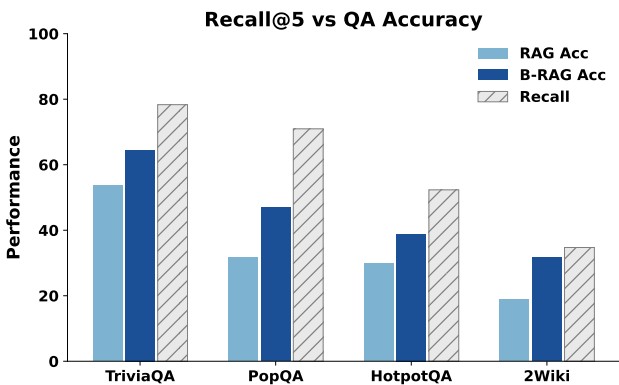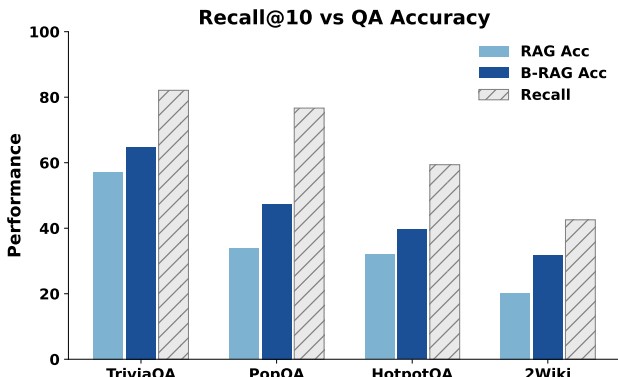

*Figure 2.* Recall@5 and Recall@10 vs. QA Accuracy across different datasets. Higher retrieval recall does not guarantee higher QA accuracy. Our method narrows the gap between recall and accuracy.

but incorrect answers. This fragility reveals a fundamental limitation: retrievers are optimized for relevance, not for providing evidence that maximally strengthens the generator's reasoning.

Current retrievers optimize exclusively for query-document relevance (Wang et al., 2022; Zhang et al., 2025), creating two systematic failure modes: they prefer trivial, answer-revealing passages that encourage shortcut learning, and they cannot distinguish genuinely unsolvable evidence from challenging yet sufficient evidence—precisely the kind that best strengthens generator reasoning. Crucially, existing retrievers operate without any estimation of the generator's competence, creating a severe train–test mismatch: systems trained on curated evidence face noisy, incomplete retrieval at deployment, leading to substantial performance degradation (Sun et al., 2025; Yu et al., 2024). Empirically, this manifests as a persistent gap between retrieval recall and end-to-end QA accuracy (Figure 2).

To overcome this limitation, we revisit the role of the reranker in RAG systems. Rather than treating it as a passive relevance scorer, we view the reranker as an active evidence set selector, responsible for choosing combinations of documents whose joint structure best supports the generator's learning and reasoning. Figure 1 contrasts this paradigm with standard relevance-based reranking. Crucially, evidence sets with similar relevance can induce drastically different learning signals: overly explicit evidence encourages shortcut learning, while incomplete evidence is fundamentally unlearnable. In contrast, evidence that is challenging yet sufficient for inference forces the generator to integrate information and resolve uncertainty.

Building on this perspective, we propose BAR-RAG, which instantiates the reranker-as-selector paradigm through boundary-aware evidence design. Rather than maximizing relevance, BAR-RAG explicitly targets the generator's Goldilocks Zone—evidence sets that are neither trivial nor

unsolvable, but lie near the generator's uncertainty boundary. We operationalize this objective via a two-stage reinforcement learning pipeline: we first train the selector to explore diverse document combinations, guided by rewards based on generator uncertainty and task success, while keeping the generator fixed; we then freeze the selector and fine-tune the generator on the induced evidence distribution, ensuring robustness under noisy retrieval.

We evaluate BAR-RAG on a diverse set of knowledge-intensive question answering benchmarks under realistic retrieval settings. Our results demonstrate that boundary-aware evidence selection consistently improves end-to-end QA performance, yields a significant performance gain of an average 10.3% over baseline models. Beyond accuracy gains, we show that BAR-RAG reshapes the evidence difficulty distribution toward the generator's competence boundary, leading to more effective learning signals and substantially improved robustness compared to relevance-based retrieval and reranking baselines.

## 2. Method

### 2.1. Overview

Our approach consists of a two-stage reinforcement learning pipeline (Figure 3). In Stage 1, we train a selector to identify evidence sets that lie within the generator's "Goldilocks Zone"—challenging enough to require genuine reasoning, yet solvable given the generator's current competence. In Stage 2, we freeze the selector and fine-tune the generator under the induced evidence distribution, thereby closing the train–test gap that plagues standard RAG systems. We describe each stage below, and introduce an iterative training scheme that progressively refines evidence selection to better match the generator's evolving competence. We describe each stage below.

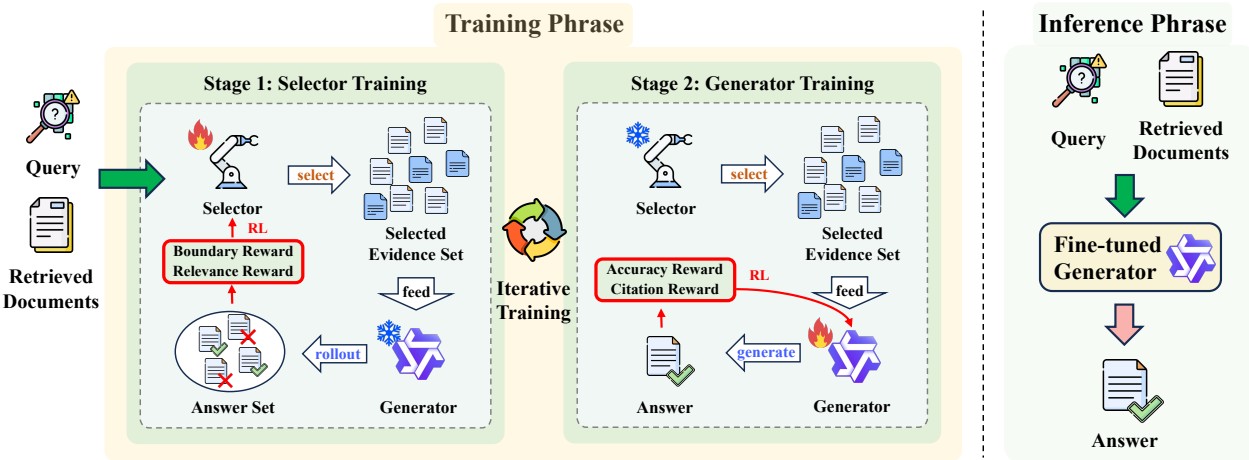

*Figure 3.* Overview of the `BAR-RAG` training and inference pipeline. During training, we adopt a two-stage framework: (Stage 1) a selector is trained with reinforcement learning using relevance and uncertainty rewards to identify challenging yet solvable evidence sets, and (Stage 2) the generator is optimized under the induced evidence distribution using accuracy, formatting, and citation rewards. At inference time, the trained generator answers questions using retrieved documents, producing robust and high-quality answers.

## 2.2. Problem Setup

Let $\pi_r$ denote the selector policy and $\pi_g$ the generator policy. Given a query $q$, let $C = \{c_1, \ldots, c_n\}$ be the top-$n$ candidate documents returned by an initial retriever (e.g., BM25 (Robertson & Zaragoza, 2009) or a dense retriever). The selector chooses a subset $S = \{c_{i_1}, \ldots, c_{i_k}\} \subseteq C$ with $k < n$, and the generator produces an answer $a$ conditioned on $(q, S)$.

Our goal is twofold. First, we train the selector policy $\pi_r$ to select evidence sets that maximize the generator's reasoning performance, not by providing trivial shortcuts, but by identifying evidence that is challenging yet solvable. Second, we train the generator policy $\pi_g$ to produce high-quality, accurate answers conditioned on the evidence sets selected by $\pi_r$, enabling robust reasoning under realistic and challenging retrieval.

**Training-time Filtering.** To ensure informative reinforcement learning signals for selector optimization, we apply a lightweight training-time filtering step that removes queries that are either trivially solvable or fundamentally unanswerable under the retrieved document pool. The key intuition is that selector learning relies on variance in generator outcomes: queries that are always answered correctly provide no incentive for evidence selection, while queries that are consistently unsolvable yield uniformly low rewards regardless of the selected evidence. We characterize trivial and unanswerable instances based on the empirical behavior of the generator over multiple rollouts given retrieved documents, using correctness statistics as a proxy for solvability. This design makes the filtering procedure domain-agnostic and inexpensive, as it depends only on generator feedback rather than task-specific heuristics or additional supervision.

The filtering is applied only during selector training to improve reward signal quality; all queries are retained during evaluation. Full details are provided in Appendix A.6.

## 2.3. Stage 1: Boundary-aware Selector Training

Standard rerankers optimize for relevance alone, often surfacing answer-revealing passages that encourage shortcut learning. In contrast, we train the selector to target the generator's competence boundary—the region where evidence is neither trivially easy ($\hat{p} \approx 1$) nor impossibly hard ($\hat{p} \approx 0$), but lies near a target difficulty level.

**The Goldilocks Zone.** Let $\hat{p}(S)$ denote the empirical probability that the generator answers correctly given evidence set $S$. We define the Goldilocks Zone as evidence sets satisfying $\hat{p}(S) \approx c$, where $c \in (0, 1)$ is a target correctness rate (e.g., $c = 0.5$). Intuitively, such evidence sets are:
(1) **Solvable**: The generator can produce correct answers with non-negligible probability, indicating that sufficient information is present.

(2) **Non-trivial**: The generator does not succeed deterministically, suggesting that reasoning is required rather than simple pattern matching.

By targeting this zone, we encourage the selector to find evidence combinations that demand genuine multi-step reasoning while remaining within the generator's capability.

**Selector sampling.** For each query $q$, we sample $M$ candidate evidence sets from the selector:

$$S^{(m)} \sim \pi_r(\cdot \mid q, C), \quad m = 1, \ldots, M. \quad (1)$$

Each evidence set $S^{(m)}$ is then evaluated using the frozen generator $\pi_g$.

**Generator rollouts.** For each evidence set $S$, we sample $K$ answers from the generator:

$$a^{(k)} \sim \pi_g(\cdot \mid q, S), \quad k = 1, \ldots, K. \quad (2)$$

Each answer must follow a prescribed format (e.g., <answer>...</answer>). We define the rollout correctness indicator based on the generator's final reward $R_g(\cdot)$ (defined in Section 2.4). Specifically, a rollout is considered correct if its reward exceeds a threshold $\delta$:

$$z^{(k)} = \mathbb{I}\left[R_g\left(a^{(k)}\right) \geq \delta\right]. \quad (3)$$

We then estimate the empirical correctness probability as:

$$\hat{p}(S) = \frac{1}{K} \sum_{k=1}^{K} z^{(k)}. \quad (4)$$

**Reward design.** We design four reward components that together guide the selector toward the Goldilocks Zone while maintaining relevance and output quality.

**(1) Boundary reward $R_{\text{bdy}}(S)$.** This reward encourages evidence sets near the target correctness $c$. We adopt a triangular function that peaks at $\hat{p}(S) = c$ and decreases linearly as the evidence becomes too easy or too hard:

$$R_{\text{bdy}}(S) = \min\left(\frac{\hat{p}(S)}{c}, \frac{1 - \hat{p}(S)}{1 - c}\right) \quad (5)$$

To estimate $\hat{p}(S)$, we use the rollout-based definition in the Generator rollouts paragraph with threshold $\delta$. A rollout is deemed correct if its generator reward $R_g(a)$ (defined in Section 2.4) exceeds a threshold $\delta$:

$$\hat{p}(S) = \frac{1}{K} \sum_{k=1}^{K} \mathbb{I}\left[R_g(a^{(k)}) \geq \delta\right]. \quad (6)$$

**(2) Relevance reward $R_{\text{rel}}(S)$.** We define the relevance reward as the average retrieval score (in the initial retrieval phrase) of the selected documents,

$$R_{\text{rel}}(S) = \frac{1}{|S|} \sum_{c_i \in S} \text{score}(q, c_i), \quad (7)$$

and rescale it to $(0, 1)$ for stability.

**(3) Format reward $R_{\text{fmt}}(S)$.** We define a binary format indicator that checks whether the selector output is well-formed (e.g., valid document indices, no duplicates, and proper structure):

$$R_{\text{fmt}}(S) = \mathbb{I}\left[S \text{ is well-formed}\right]. \quad (8)$$

This design ensures that malformed selector outputs receive zero reward, while relative trade-offs among boundary targeting, relevance, and document count are only considered for valid evidence sets.

**(4) Count penalty $P_{\text{cnt}}(S)$.** To encourage the selector to output a target number of documents $k^*$, we apply a penalty proportional to the deviation:

$$P_{\text{cnt}}(S) = \min\left(\alpha \cdot \lVert S \rVert - k^* \rvert, P_{\text{max}}\right), \quad (9)$$

where $\alpha$ is the penalty per document deviation and $P_{\text{max}}$ caps the maximum penalty.

**Final reward.** We combine the reward components using the format indicator as a gate:

$$R_r(S) = R_{\text{fmt}}(S) \cdot \left(\lambda_{\text{bdy}} R_{\text{bdy}}(S) + \lambda_{\text{rel}} R_{\text{rel}}(S) - P_{\text{cnt}}(S)\right), \quad (10)$$

where $\lambda_{\text{bdy}}, \lambda_{\text{rel}} \geq 0$ control the relative importance of targeting the competence boundary versus maintaining relevance.

**Optimization.** We optimize the selector using Group Relative Policy Optimization (GRPO) (Shao et al., 2024). For each group of sampled evidence sets $\{S^{(m)}\}_{m=1}^{M}$, we compute group-normalized advantages:

$$\mathcal{A}^{(m)} = \frac{R_r(S^{(m)}) - \mu}{\sigma + \epsilon}, \quad (11)$$

where $\mu$ and $\sigma$ are the mean and standard deviation of rewards within the group. The selector is trained to minimize:

$$\mathcal{L}_r(\theta_r) = -\mathbb{E}_m\Big[\min\Big(r^{(m)}\mathcal{A}^{(m)},$$
$$\text{clip}(r^{(m)}, 1 - \epsilon, 1 + \epsilon)\mathcal{A}^{(m)}\Big)\Big], \quad (12)$$

where $r^{(m)} = \pi_r(S^{(m)} \mid q, C)/\pi_r^{\text{old}}(S^{(m)} \mid q, C)$ is the likelihood ratio.

## 2.4. Stage 2: Generator Fine-tuning

Standard RAG pipelines train generators on curated, near-perfect evidence but deploy them with noisy retrieval, leading to a mismatch and performance degradation. By fine-tuning the generator under the selector's output distribution, we expose it to realistic, challenging evidence during training.

**Training procedure.** For each query $q$, the frozen selector produces an evidence set $S = \pi_r(q, C)$. The generator samples $K$ candidate answers:

$$a^{(k)} \sim \pi_g(\cdot \mid q, S), \quad k = 1, \ldots, K. \quad (13)$$

*Table 1.* **BAR-RAG consistently improves performance across models and QA benchmarks.** Exact match (EM) results on general QA and multi-hop QA tasks for three backbone models. All values are reported as absolute scores. For IRCoT, RAG w/ Reranker, RAG SFT, and BAR-RAG, cell shading indicates improvement ( blue ) or degradation ( red ) relative to the RAG baseline. BAR-RAG yields strong and consistent gains across both single-hop and multi-hop settings, surpassing standard RAG pipelines and reranking-based baselines.

| | General QA | | | Multi-Hop QA | | | | |
|---|---|---|---|---|---|---|---|---|
| **Method** | **NQ** | **TriviaQA** | **PopQA** | **HotpotQA** | **2Wiki.** | **MuSiQue** | **Bamboogle** | **Avg.** |
| **Qwen-2.5-3B-Instruct** | | | | | | | | |
| Direct Inference | 10.6 | 28.8 | 10.8 | 14.9 | 24.4 | 2.0 | 2.4 | 13.4 |
| CoT | 2.3 | 3.2 | 0.5 | 2.1 | 2.1 | 0.2 | 0.0 | 1.5 |
| RAG | 34.8 | 54.4 | 38.7 | 25.5 | 22.6 | 4.7 | 8.0 | 27.0 |
| IRCoT | 11.1 | 31.2 | 20.0 | 16.4 | 17.1 | 6.7 | 24.0 | 18.1 |
| RAG w/Reranker | 35.6 | 55.3 | 39.6 | 26.7 | 23.4 | 4.9 | 10.4 | 28.0 |
| RAG SFT | 38.9 | 58.1 | 41.4 | 28.9 | 25.9 | 6.4 | 13.5 | 30.4 |
| **BAR-RAG** (1 Iter) | 40.1 | 58.4 | 41.2 | 31.0 | 24.9 | 6.3 | 24.0 | 32.3 |
| **BAR-RAG** (2 Iter) | 41.5 | 61.3 | 43.4 | 33.2 | 26.4 | 7.4 | 26.3 | 34.3 |
| **BAR-RAG** (3 Iter) | 42.0 | 59.7 | 44.1 | 32.6 | 26.7 | 7.2 | 26.8 | 34.2 |
| **Qwen-2.5-7B-Instruct** | | | | | | | | |
| Direct Inference | 13.4 | 40.8 | 14.0 | 18.3 | 12.6 | 3.1 | 12.0 | 16.3 |
| CoT | 4.8 | 18.5 | 5.4 | 9.2 | 10.8 | 2.2 | 23.2 | 10.6 |
| RAG | 39.3 | 53.7 | 26.7 | 28.9 | 18.9 | 4.7 | 16.0 | 26.9 |
| IRCoT | 22.4 | 47.8 | 30.1 | 13.3 | 14.9 | 7.2 | 22.4 | 22.6 |
| RAG w/Reranker | 40.5 | 55.3 | 27.3 | 28.1 | 20.4 | 5.5 | 18.7 | 28.0 |
| RAG SFT | 42.7 | 58.6 | 32.3 | 32.4 | 22.6 | 6.8 | 27.1 | 31.8 |
| **BAR-RAG** (1 Iter) | 44.7 | 63.2 | 44.1 | 37.2 | 28.3 | 8.3 | 36.7 | 37.2 |
| **BAR-RAG** (2 Iter) | 46.1 | 64.3 | 46.3 | 38.1 | 29.9 | 9.1 | 39.1 | 38.8 |
| **BAR-RAG** (3 Iter) | 46.9 | 64.5 | 46.9 | 38.8 | 29.8 | 9.1 | 39.6 | 39.1 |
| **LLaMA-3.1-8B-Instruct** | | | | | | | | |
| Direct Inference | 18.4 | 36.5 | 19.8 | 12.5 | 23.0 | 2.7 | 8.8 | 17.4 |
| CoT | 27.8 | 54.1 | 23.5 | 24.1 | 23.0 | 6.8 | 16.3 | 25.1 |
| RAG | 42.7 | 58.2 | 28.9 | 30.3 | 19.4 | 6.3 | 17.6 | 29.1 |
| IRCoT | 23.5 | 48.1 | 31.2 | 12.2 | 11.8 | 6.9 | 24.5 | 22.6 |
| RAG w/Reranker | 43.6 | 60.1 | 30.4 | 32.5 | 23.6 | 7.6 | 19.5 | 31.0 |
| RAG SFT | 45.8 | 63.4 | 35.6 | 35.9 | 27.4 | 8.9 | 24.3 | 34.5 |
| **BAR-RAG** (1 Iter) | 47.5 | 65.9 | 45.8 | 39.5 | 31.4 | 11.1 | 30.4 | 38.8 |
| **BAR-RAG** (2 Iter) | 49.0 | 67.7 | 48.0 | 40.5 | 33.0 | 12.5 | 32.8 | 40.5 |
| **BAR-RAG** (3 Iter) | 49.5 | 67.3 | 48.6 | 41.2 | 33.0 | 12.0 | 33.3 | 40.7 |

**Generator Reward Design.** We design a composite reward that encourages both answer accuracy and proper evidence attribution.

**(1) Format reward.** The generator must produce outputs in the prescribed format (i.e., valid <answer> tags). If the format check fails, the reward is set to zero:

$$R_g(a) = 0 \quad \text{if format is invalid.} \quad (14)$$

**(2) Accuracy reward** $R_{\text{acc}}(a)$**.** For well-formed outputs, we compute a weighted combination of token-level F1 score and exact match (EM) against the gold answers:

$$R_{\text{acc}}(a) = \beta_1 \cdot \max_{g \in \mathcal{G}} F_1(a, g) + \beta_2 \cdot \max_{g \in \mathcal{G}} \text{EM}(a, g), \quad (15)$$

where $\mathcal{G}$ is the set of gold answers, and $\beta_1, \beta_2 \geq 0$ control the relative importance of partial credit (F1) versus exact correctness (EM).

**(3) Citation reward** $R_{\text{cite}}(a)$**.** To encourage the generator to ground its reasoning in the provided evidence, we reward appropriate citation behavior in the <think> block. Let $n_{\text{cite}}$ denote the number of unique documents cited. We use a peaked reward centered at a target citation count $n^*$:

$$R_{\text{cite}}(a) = \begin{cases} 1.0, & \text{if } n_{\text{cite}} = n^*, \\ 0.5, & \text{if } |n_{\text{cite}} - n^*| = 1, \\ 0.0, & \text{otherwise.} \end{cases} \quad (16)$$

This encourages the generator to cite a moderate number of sources from the provided documents, sufficient to support

Table 2. Ablation study on different components.

| Method | NQ | PopQA | HotpotQA | Avg. |
|---|---|---|---|---|
| Full | 46.9 | 46.9 | 38.8 | 44.2 |
| w/o Filtering | 42.6 | 32.1 | 32.3 | 35.6 |
| w/o Stage1 | 42.1 | 42.5 | 41.9 | 42.2 |
| w/o Stage2 | 39.7 | 34.1 | 36.7 | 36.8 |
| *Selector* | | | | |
| w/o $R_{\text{bdy}}$ | 43.4 | 43.6 | 33.5 | 40.2 |
| w/o $R_{\text{rel}}$ | 46.1 | 46.3 | 38.1 | 43.5 |
| *Generator* | | | | |
| w/o $R_{\text{cite}}$ | 45.3 | 44.8 | 37.9 | 42.7 |

multi-hop reasoning but not so many as to dilute focus.

**Final reward.** We combine the three components with independent weights for accuracy and citation, while fixing the format coefficient:

$$R_g(a) = R_{\text{fmt}}(a) \cdot \big(\lambda_{\text{acc}} R_{\text{acc}}(a) + \lambda_{\text{cite}} R_{\text{cite}}(a)\big), \quad (17)$$

where $\lambda_{\text{acc}}, \lambda_{\text{cite}} \geq 0$ are hyperparameters.

**Optimization.** The generator is optimized using GRPO with the same clipped objective:

$$\mathcal{L}_g(\theta_g) = -\mathbb{E}_k \Big[ \min \Big( r^{(k)} \mathcal{A}^{(k)}, \\ \text{clip}(r^{(k)}, 1 - \epsilon, 1 + \epsilon) \mathcal{A}^{(k)} \Big) \Big], \quad (18)$$

where advantages $\mathcal{A}^{(k)}$ are computed from generator rewards within each group, and $r^{(k)} = \pi_g(a^{(k)} \mid q, S)/\pi_g^{\text{old}}(a^{(k)} \mid q, S)$.

**Iterative two-stage training.** Rather than a single pass, we alternate between the two stages for $T$ iterations. At iteration $t$, we (i) train the selector $\pi_r^{(t)}$ against a frozen generator $\pi_g^{(t-1)}$ to target the current competence boundary, and then (ii) freeze the updated selector and fine-tune the generator to obtain $\pi_g^{(t)}$ under the induced evidence distribution. In practice, each iteration consists of one epoch of selector training followed by one epoch of generator fine-tuning. This alternating procedure progressively refines evidence selection to track the generator's evolving competence. Algorithmic details are provided in Appendix 1.

## 2.5. Inference

The selector is used only during training to shape a challenging evidence distribution for the generator. At inference time, we discard the selector entirely and apply the fine-tuned generator directly to the top-$k$ documents returned by a standard retriever (e.g., BM25 or a dense retriever), producing the final answer $a = \pi_g(q, S)$. This design incurs no

additional inference cost and preserves the standard RAG pipeline. By being trained iteratively on adversarially selected evidence near its competence boundary, the generator becomes more robust to noisy, incomplete, and imperfect retrieval results encountered at test time.

## 3. Experiments

### 3.1. Datasets and Evaluation Metrics

We evaluate on seven knowledge-intensive QA datasets spanning diverse reasoning challenges. For **single-hop QA**, we use Natural Questions (NQ) (Kwiatkowski et al., 2019), TriviaQA (Joshi et al., 2017), and PopQA (Mallen et al., 2023), which test robustness to retrieval noise, paraphrased evidence, and long-tail entity reasoning, respectively. For **multi-hop QA**, we use HotpotQA (2-hop bridge reasoning) (Yang et al., 2018), 2WikiMultiHopQA (distant supporting facts) (Ho et al., 2020), MuSiQue (3–5 compositional hops) (Trivedi et al., 2022), and Bamboogle (indirect reasoning) (Press et al., 2023). The training datasets are reported at Appendix A.7. We report Exact Match (EM) as the primary metric, following standard evaluation protocols.

### 3.2. Baselines

We compare against two categories of methods. **Without retrieval**: (1) Direct Inference, prompting the base model directly; (2) Chain-of-Thought (CoT) (Wei et al., 2022), eliciting step-by-step reasoning; (3) RAG SFT, supervised fine-tuning on QA pairs with retrieved evidence. **With retrieval**: (1) RAG (Shi et al., 2024), standard dense retrieval-augmented generation; (2) RAG w/ Reranker, adding a neural reranker to re-score retrieved documents before generation; (3) IRCoT (Trivedi et al., 2023), a multi-step QA framework that interleaves retrieval with steps in a CoT; (4) RAG SFT, fine-tuning on QA pairs augmented with top-5 retrieved passages, exposing the model to both relevant and noisy evidence during training. All baselines use the same base model (*Qwen2.5-3B-Instruct*, *Qwen2.5-7B-Instruct* (Qwen et al., 2025) and *LLaMA-3.1-8B-Instruct* (Grattafiori et al., 2024)), retriever (*E5-base-v2* (Wang et al., 2022)) and reranker (*Qwen-3-Embedding-8B* (Zhang et al., 2025)) and Top-5 retrieved documents as input for fair comparison.

### 3.3. Implementation Details

We use instruction-tuned LLMs as both the selector and generator backbones, including Qwen2.5 and LLaMA-3.1 variants. Detailed model configurations, reward parameters, rollout settings, and training schedules are provided in Appendix A.5.

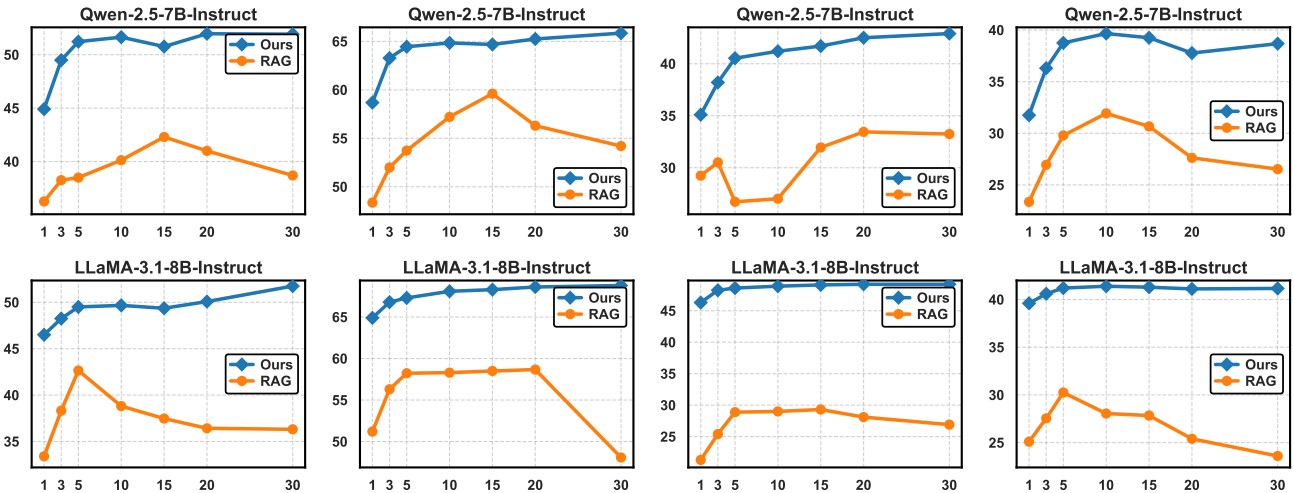

**Figure 4.** Top-$K$ accuracy curves on four QA benchmarks for two base models: *Qwen-2.5-7B-Instruct* and *LLaMA-3.1-8B-Instruct*. From left to right, columns correspond to **NQ**, **TriviaQA**, **PopQA**, and **HotpotQA**. Across both models and all datasets, our method consistently achieves higher accuracy in low-$K$ regimes and remains robust as $K$ increases, whereas standard RAG exhibits weaker scaling behavior and higher sensitivity to retrieval noise.

### 3.4. Main Results

Table 1 presents our main results across seven QA benchmarks. `BAR-RAG` consistently outperforms all baselines on both general and multi-hop QA tasks. Using *Qwen2.5-7B-Instruct* as a representative example, `BAR-RAG` achieves substantial improvements over the strongest baseline RAG SFT on single-hop benchmarks: +8.5 on NQ (51.2 vs. 42.7), +5.9 on TriviaQA, and +14.6 on PopQA. The advantages are more pronounced on multi-hop tasks, with gains of +6.4 on HotpotQA, +7.2 on 2WikiMultiHopQA, and +12.5 on Bamboogle—the latter highlighting the benefit of training on evidence that demands genuine multi-step reasoning. Results on *LLaMA-3.1-8B-Instruct* and *Qwen2.5-3B-Instruct* confirm that our approach generalizes across model families, achieving average EM of 40.7 and 34.2 respectively (vs. 29.1 and 27.0 for RAG).

**Iterative Improvement**  We examine how performance evolves across training iterations. As shown in Table 1, all three models exhibit consistent improvement from Iter 1 to Iter 2, with diminishing gains from Iter 2 to Iter 3. This pattern suggests that the iterative co-training procedure converges toward a stable solution within a few iterations, and that the majority of gains are captured in the first two rounds of selector-generator co-adaptation.

### 3.5. Ablation Studies

We conduct ablation studies to validate the necessity of key design choices, including (i) the design of reward components for boundary-aware evidence selection and (ii) the two-stage training pipeline.

**Effect of Reward Components.**  To examine the contribution of individual reward terms in the selector, we ablate two key components: the boundary reward $R_{\text{bdy}}$ and the relevance reward $R_{\text{rel}}$, while keeping other training settings unchanged. Specifically, we evaluate variants that remove $R_{\text{bdy}}$ or $R_{\text{rel}}$ from the selector training objective. Results are reported in Table 2. Removing the boundary reward leads to a clear performance degradation across all datasets, confirming that targeting evidence near the generator's competence boundary is crucial for robust reasoning. In contrast, removing the relevance reward results in a milder but consistent drop, indicating its complementary role in maintaining evidence quality. We further ablate the citation reward $R_{\text{cite}}$ used in generator fine-tuning. Removing $R_{\text{cite}}$ results in a consistent but smaller performance drop, indicating that citation supervision complements competence-aware evidence selection by improving answer grounding rather than driving the main gains.

**Effect of Two-Stage Training.**  Our pipeline consists of two stages: selector training (Stage 1) followed by generator fine-tuning (Stage 2). To assess the necessity of each component, we conduct ablation studies summarized in Table 2. When Stage 2 is removed, the selector is trained while keeping the generator frozen at its initial state. This variant leads to a clear performance drop across all benchmarks, indicating that adapting the generator to the competence-boundary evidence distribution is crucial for effective reasoning. We further examine the role of the training-time filtering step by removing it and training the selector on the full dataset. Eliminating filtering consistently degrades performance, with particularly large drops on PopQA and HotpotQA, confirming that trivially solvable and unanswer-

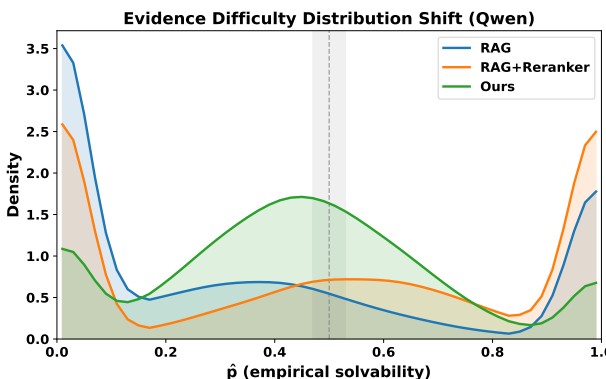 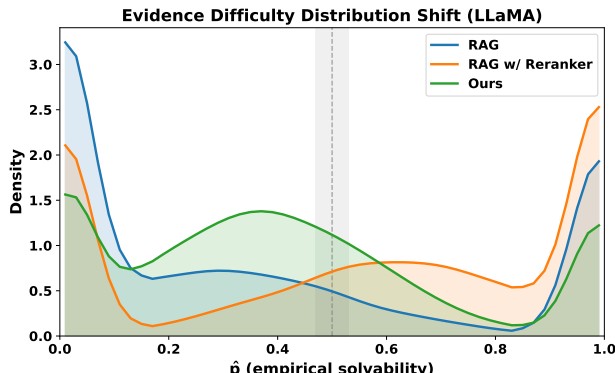

*Figure 5.* Distribution shift of evidence difficulty ($\hat{p}$) across different retrieval methods. Our method concentrates samples toward the target correctness level $c$=0.5, substantially reducing extreme cases near 0 (unsolvable) and 1 (trivially solvable).

able queries introduce degenerate RL signals that hinder stable selector optimization. Finally, removing Stage 1 eliminates competence-aware evidence selection altogether, reducing the pipeline to standard retrieval followed by generator training. The resulting degradation demonstrates that naive retrieval is insufficient for robust multi-hop reasoning.

### 3.6. Analysis

**Generator Robustness to Evidence Quality.** A central claim of our approach is that training on competence-boundary evidence yields a generator that is inherently more robust to evidence quality—even when the selector is discarded at inference time. To verify this, we evaluate generators trained with BAR-RAG, and standard RAG, using the same naive retriever outputs (top-$k$ by retrieval score) under varying evidence budgets $k \in \{1, 3, 5, 10, 15, 30\}$. We conduct this evaluation across two backbone models (*Qwen-2.5-7B-Instruct* and *LLaMA-3.1-8B-Instruct*) and four datasets (NQ, TriviaQA, PopQA, and HotpotQA). Figure 4 reports K-accuracy curves for all settings. Consistent with our hypothesis, BAR-RAG-trained generators consistently outperform baselines across all $k$, with the largest gains appearing in low-$k$ regimes where evidence is sparse and reasoning robustness is most critical.

**Evidence Difficulty Distribution Shift.** For each question, we estimate the empirical solvability $\hat{p}(S)$ of an evidence set $S$ via $K$ generator rollouts, and compare the resulting distributions across three methods: Naive RAG (selecting top-$k$ documents by retrieval score), RAG with a neural reranker, and our BAR-RAG selector. Figure 5 compares the resulting distributions across Naive RAG, RAG with a neural reranker, and our BAR-RAG selector for both *Qwen-2.5-7B-Instruct* and *LLaMA-3.1-8B-Instruct*. While Naive RAG and reranker-based RAG exhibit bimodal distributions dominated by unsolvable or trivially answer-revealing contexts, BAR-RAG suppresses both extremes and

concentrates probability mass around the target correctness level $c = 0.5$. This consistent shift across generators indicates that BAR-RAG actively promotes hard-but-solvable evidence near the generator's competence boundary rather than merely improving relevance.

Additional analyses, including counterfactual evidence-dependence studies and comparisons with agentic RAG methods, are provided in Appendix A.3 and A.2.

## 4. Related Work

**Retrieval-Augmented Generation** Retrieval-Augmented Generation (RAG) improves factual accuracy by grounding language model outputs in external knowledge sources (Lewis et al., 2020; Guu et al., 2020). Recent work explores tighter retrieval–generation integration, including mechanisms that adapt retrieval behavior based on model uncertainty or self-reflection, such as Self-RAG (Asai et al., 2024). However, BAR-RAG targets what evidence to retrieve, framing evidence selection as a competence-aware process that deliberately selects hard-but-solvable contexts to strengthen the generator's reasoning ability.

**Document Reranking** Reranking refines retrieved candidates before downstream generation, with early neural approaches relying on cross-encoder architectures to compute relevance scores (Nogueira & Cho, 2019). More recently, reinforcement learning has been used to optimize reranking policies via generator feedback, as in DynamicRAG (Sun et al., 2025). In contrast to prior RL-based rerankers that optimize relevance or answer quality, BAR-RAG introduces an explicit competence boundary reward that directly models evidence difficulty, enabling the selection of challenging yet solvable evidence sets.

Further extand related works are reported in Appendix A.4.

# 5. Conclusion

We presented `BAR-RAG`, a boundary-aware evidence selection training framework that trains the selector to target hard-but-solvable evidence near the generator's competence boundary. We train generator with realistic retrieval conditions, yielding consistent robustness gains across diverse QA benchmarks without additional inference-time cost.

# Impact Statement

This paper presents work whose goal is to advance the field of machine learning. There are many potential societal consequences of our work, none of which we feel must be specifically highlighted here.

# Acknowledgement

Research was supported in part by the AI Institute for Molecular Discovery, Synthetic Strategy, and Manufacturing: Molecule Maker Lab Institute (MMLI), funded by U.S. National Science Foundation under Award 2505932, NSF IIS 25-37827, and the Institute for Geospatial Understanding through an Integrative Discovery Environment (I-GUIDE) by NSF under Award No. 2118329. Any opinions, findings, and conclusions or recommendations expressed herein are those of the authors and do not necessarily represent the views, either expressed or implied, of DARPA or the U.S. Government.

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

# A. Appendix

## A.1. Algorithm

We present our algorithm of CompRAG in Algorithm 1.

---

**Algorithm 1** Boundary-aware Evidence Selection Training (BAR-RAG)

---

**Require:** Training dataset $\mathcal{D}$, retriever $\mathcal{R}$, selector $\pi_r$, generator $\pi_g$
**Require:** Target correctness $c$, reward threshold $\delta$, rollouts $K$, evidence samples $M$
**Require:** Training iterations $T$
 1: **// Training-time filtering**
 2: $\mathcal{D}_{\text{filt}} \leftarrow \text{FILTER}(\mathcal{D})$
 3: **for** $t = 1$ to $T$ **do**
 4:     **// Stage 1: Selector Training (generator frozen)**
 5:     **for** each query $q$ in batch from $\mathcal{D}_{\text{filt}}$ **do**
 6:         $C \leftarrow \mathcal{R}(q)$
 7:         Sample $M$ evidence sets $\{S^{(m)}\}_{m=1}^{M}$ from $\pi_r$
 8:         **for** each $S^{(m)}$ **do**
 9:             Sample $K$ answers from $\pi_g(\cdot \mid q, S^{(m)})$
10:             Estimate $\hat{p}(S^{(m)})$ and compute selector reward $R_r(S^{(m)})$
11:         **end for**
12:         Update selector $\pi_r$ via GRPO
13:     **end for**
14:     **// Stage 2: Generator Training (selector frozen)**
15:     **for** each query $q$ in batch from $\mathcal{D}$ **do**
16:         $C \leftarrow \mathcal{R}(q)$
17:         $S \leftarrow \pi_r(q, C)$
18:         Sample $K$ answers from $\pi_g(\cdot \mid q, S)$
19:         Update generator $\pi_g$ via GRPO
20:     **end for**
21: **end for**

---

## A.2. Comparison with Recent Reasoning and Agentic RAG Methods

To further contextualize BAR-RAG within the rapidly evolving landscape of reasoning-centric and agentic RAG systems, we compare against several recent representative methods, including Search-o1 (Li et al., 2025a), Search-R1 (Jin et al., 2025), and DynamicRAG (Sun et al., 2025). [1] These approaches emphasize explicit multi-step search, iterative retrieval, or reinforcement-learning-based reranking at inference time, and are often evaluated under substantially different retrieval and search budgets.

In particular, we group results by backbone scale following the structure of Table 3, and report the best-performing iteration of BAR-RAG for each backbone. Importantly, BAR-RAG incurs no additional inference-time overhead beyond standard RAG, whereas the compared reasoning and agentic methods typically rely on multi-round search or tool invocation during inference.

## A.3. Counterfactual Evidence Dependence Analysis

A potential concern with generator-aware evidence selection is whether performance gains arise from genuine evidence use or from self-confirmation effects induced by generator feedback. To directly test whether the model's predictions causally depend on the evidence it cites, we perform a counterfactual evidence-dependence analysis.

For each test example, we first run the model under the standard setting (FULL) using the top-$k$ retrieved documents, and

---

[1] As these methods vary widely in backbone models, inference-time computation, and retrieval interfaces, a strictly controlled apples-to-apples comparison is not always possible. Therefore, this table is intended to provide a qualitative and contextual comparison rather than a claim of direct superiority.

*Table 3.* Comparison with recent reasoning and agentic RAG methods (EM, %). Results are grouped by backbone following Table 1, and BAR-RAG reports the best iteration for each backbone. Search-o1 and Search-R1 results are taken directly from the original papers. DynamicRAG results are obtained by re-running the authors' released checkpoint under our evaluation pipeline, as its original setting is not fully aligned with ours. Notably, Search-R1 employs substantially different retrieval configurations on multi-hop QA (HotpotQA, 2WikiMultiHopQA, MuSiQue and Bamboogle), involving multiple rounds of search with multiple documents per round, whereas our setting uses a single-shot top-$k$ retrieval (top-5) without iterative search. As a result, Search-R1 results on these datasets (marked with [†]) are not directly comparable to `BAR-RAG` and should be interpreted with caution.

| Method | NQ | TriviaQA | PopQA | HotpotQA | 2Wiki | MuSiQue | Bamboogle | Avg. |
|---|---|---|---|---|---|---|---|---|
| **Qwen2.5-3B-Instruct** | | | | | | | | |
| Search-o1 | 23.8 | 47.2 | 26.2 | 22.1 | 21.8 | 5.4 | 32.0 | 25.5 |
| Search-R1 | 34.1 | 54.5 | 37.8 | 32.4† | **31.9†** | **10.3†** | **26.4†** | 32.5 |
| **BAR-RAG** | **41.5** | **61.3** | **43.4** | **33.2** | 26.4 | 7.4 | 26.3 | **34.3** |
| **Qwen2.5-7B-Instruct** | | | | | | | | |
| Search-o1 | 15.1 | 44.3 | 13.1 | 18.7 | 17.6 | 5.8 | 29.6 | 20.6 |
| Search-R1 | 39.3 | 61.0 | 39.7 | 37.0† | **41.4†** | **14.6†** | 36.8† | 38.5 |
| **BAR-RAG** | **46.9** | **64.5** | **46.9** | **38.8** | 29.8 | 9.1 | **39.6** | **39.1** |
| **LLaMA-3.1-8B-Instruct** | | | | | | | | |
| DynamicRAG | 46.4 | 57.5 | 36.7 | 34.2 | – | – | – | |
| **BAR-RAG** | **49.5** | **67.3** | **48.6** | **41.2** | 33.0 | 12.0 | 33.3 | 40.7 |

record the set of documents cited in the model's generation. We then construct two counterfactual variants while keeping the question and decoding settings fixed: (i) REMOVE-CITED, where all documents cited by the model in the original generation are removed from the input, and (ii) KEEP-ONLY-CITED, where only the cited documents are retained and all other retrieved documents are discarded. The cited document set is always determined from the original FULL generation and kept fixed across counterfactual runs.

Table 4 reports Exact Match (EM) results on NQ, HotpotQA, and Bamboogle using Qwen2.5-7B-Instruct. Across all datasets, removing the cited evidence causes a large performance drop, with $\Delta_{\mathrm{rm}}$ exceeding 24 EM on all three benchmarks. This sharp degradation indicates that correct predictions critically rely on the documents explicitly referenced by the model.

In contrast, retaining only the cited documents largely preserves performance and in some cases slightly improves it (e.g., NQ and Bamboogle), suggesting that the cited documents form a compact and sufficient support set, while additional retrieved documents often act as noise.

## A.4. Extended Related Work

### A.4.1. RETRIEVAL-AUGMENTED GENERATION

Retrieval-Augmented Generation (RAG) has emerged as a general framework for grounding language model outputs in external knowledge sourceAR s, mitigating hallucination and enabling dynamic knowledge updates. Early formulations combine neural retrieval with sequence-to-sequence generation for knowledge-intensive tasks (Lewis et al., 2020; Guu et al., 2020; Wang et al., 2025; Cao et al., 2025; Huang et al., 2025; Wei et al., 2025; Sun et al., 2026a;b). Beyond document-level

*Table 4.* Counterfactual evidence-dependence analysis on Qwen2.5-7B-Instruct. REMOVE-CITED removes all documents cited by the model in the original generation, while KEEP-ONLY-CITED retains only the cited documents. $\Delta_{\mathrm{rm}}$ denotes the EM drop from FULL to REMOVE-CITED.

| Dataset | Full | Remove-Cited | Keep-Only-Cited | $\Delta_{\mathrm{rm}}$ |
|---|---|---|---|---|
| NQ | 46.9 | 20.4 | 47.3 | 26.5 |
| HotpotQA | 38.8 | 12.2 | 38.5 | 26.6 |
| Bamboogle | 39.6 | 15.3 | 40.1 | 24.3 |

retrieval, several works explore datastore-based or token-level augmentation. For example, kNN-LM (Khandelwal et al., 2020) augments next-token prediction with nearest-neighbor retrieval over a large datastore, while subsequent extensions improve efficiency and contextualization (Ram et al., 2023).

Recent research has focused on tighter integration between retrieval and generation. Some approaches train retrieval and generation components end-to-end, while others introduce explicit control mechanisms to decide when retrieval should occur. Self-RAG (Asai et al., 2024) equips the generator with a learned critic that reflects on intermediate generations and dynamically triggers retrieval. These methods primarily focus on retrieval timing or integration strategy, whereas our work complements them by addressing the orthogonal problem of evidence selection quality under fixed retrieval budgets.

### A.4.2. DOCUMENT RERANKING

Reranking is a long-standing component in information retrieval pipelines, aiming to refine initial retrieval results before downstream consumption. Traditional learning-to-rank approaches optimize pairwise or listwise objectives over hand-crafted features (Cao et al., 2007). With the advent of pretrained language models, neural rerankers based on cross-encoder architectures have become the dominant paradigm, jointly encoding query–document pairs to compute relevance scores with fine-grained token interactions (Nogueira & Cho, 2019).

Large language models have recently been explored as rerankers, leveraging their reasoning capabilities to assess document usefulness. Prior work spans multiple granularities, including pointwise relevance prediction or likelihood estimation (Drozdov et al., 2023; Sachan et al., 2022), pairwise comparison of candidate documents (Qin et al., 2024), and listwise ranking that directly outputs document permutations (Sun et al., 2023; Ma et al., 2023). Many of these approaches operate in a zero-shot or weakly supervised setting and optimize relevance-based objectives.

Several recent works incorporate reinforcement learning to optimize reranking policies. DynamicRAG (Sun et al., 2025) formulates reranking as a sequential decision process and uses generator feedback as a reward signal to dynamically determine both the ordering and the number of documents. While effective, such approaches typically rely on relevance or answer-quality-based rewards. In contrast, our method introduces an explicit competence-aware objective that directly models evidence difficulty, targeting hard-but-solvable evidence sets that maximize learning signal for downstream generator training.

### A.5. Training Implementation Details

We use *Qwen2.5-3B-Instruct*, *Qwen2.5-7B-Instruct*, and *LLaMA-3.1-8B-Instruct* as backbone models for both the selector and the generator. In both training stages, we adopt parameter-efficient fine-tuning using LoRA adapters with rank 32 and scaling factor $\alpha = 16$, while keeping all backbone parameters frozen.

For retrieval, we use *E5* as the dense retriever to obtain a fixed pool of top-$n = 25$ candidate documents for each query. During selector training, we sample $M = 8$ candidate evidence sets per query, each consisting of $k = 5$ documents.

Each sampled evidence set is evaluated using $K = 10$ generator rollouts to estimate the correctness probability $\hat{p}(S)$. A rollout is deemed correct if its final generator reward exceeds a threshold $\delta = 0.8$. The selector reward targets a correctness level of $c = 0.5$, with boundary and relevance weights $\lambda_{\text{bdy}} = 1.0$ and $\lambda_{\text{rel}} = 0.2$, respectively. We use a relevance temperature $\tau = 10.0$ and apply a count penalty of $\alpha = 0.5$ per document deviation from the target size $k^* = 5$, capped at $P_{\max} = 1.0$.

The generator reward combines answer accuracy and citation quality with weights $\lambda_{\text{acc}} = 0.8$ and $\lambda_{\text{cite}} = 0.2$. The accuracy reward weights token-level F1 and exact match as $\beta_1 = 0.7$ and $\beta_2 = 0.3$, respectively. The citation reward targets $n^* = 2$ cited documents.

Both stages are optimized using GRPO with a learning rate of $4 \times 10^{-6}$, cosine learning rate decay with 2% warmup, clipping parameter $\epsilon = 0.2$, batch size 8, and KL regularization coefficient 0.001. Training is conducted for three iterations on $8 \times$A100 (40GB) GPUs using bfloat16 mixed-precision training with gradient checkpointing.

### A.6. Training-time Filtering Details

We now provide implementation details for the filtering procedure described in Section **??**. For each training query, we sample $N$ reranker rollouts and $K$ generator rollouts per evidence set. Generator outputs are judged correct if their total reward exceeds a fixed threshold $\delta$, consistent with the correctness definition used during selector training.

*Table 5.* Dataset statistics for training and evaluation.

| Split | Dataset | #Examples | Task Type | Domain |
|-------|---------|----------:|-----------|--------|
| Train | NQ | 79,168 | Single-hop | In-domain |
|  | HotpotQA | 8,757 | Multi-hop | In-domain |
|  | *Total* | *87,925* | – | – |
| Eval | NQ | 3,610 | Single-hop | In-domain |
|  | HotpotQA | 7,405 | Multi-hop | In-domain |
|  | TriviaQA | 11,313 | Single-hop | Out-of-domain |
|  | PopQA | 14,267 | Single-hop | Out-of-domain |
|  | 2WikiMultiHopQA | 12,576 | Multi-hop | Out-of-domain |
|  | Musique | 2,417 | Multi-hop | Out-of-domain |
|  | Bamboogle | 125 | Multi-hop | Out-of-domain |

In practice, we compute the mean and variance of empirical correctness across reranker rollouts. Queries with near-zero variance or near-deterministic outcomes are removed, as they correspond to trivially easy or unanswerable cases. Unless otherwise stated, we retain queries whose mean correctness satisfies $\mu_q \in [m_{\min}, m_{\max}]$ and whose variance exceeds $v_{\min}$.

In all experiments, we use $N = 8$ reranker rollouts and $K = 10$ generator rollouts. We set the correctness threshold to $\delta = 0.5$. The mean correctness bounds are $m_{\min} = 0.25$ and $m_{\max} = 0.85$, and the minimum variance threshold is $v_{\min} = 0.02$. This filtering step is applied only to selector training and does not affect generator training.

### A.7. Training Data

Following Jin et al. (2025), we construct the training set by merging the training splits of Natural Questions (NQ) (Kwiatkowski et al., 2019) and HotpotQA (Yang et al., 2018). NQ provides single-hop factoid questions derived from real Google search queries, while HotpotQA contributes multi-hop questions that require reasoning over multiple Wikipedia passages. This combination ensures coverage of both simple retrieval scenarios and complex multi-step reasoning tasks.

For the retrieval corpus, we use the 2018 Wikipedia dump (Karpukhin et al., 2020), which contains approximately 21 million passages. We employ E5 (Wang et al., 2022) as the dense retriever. For each query, we retrieve the top-25 passages to form a fixed retrieval pool, from which the selector learns to compose evidence subsets during training.

Table 5 summarizes the dataset statistics. The combined training set contains 87,925 question-answer pairs. Each training instance consists of a question $q$, the ground-truth answer $a$, and a retrieval pool $\mathcal{D}_q$ containing the top-25 passages retrieved for $q$. During BAR-RAG training, the selector operates over this fixed retrieval pool, learning to compose evidence subsets that challenge the generator near its competence boundary.

We evaluate on seven benchmark datasets to assess both in-domain and out-of-domain generalization: (1) **In-domain**: NQ and HotpotQA; (2) **Out-of-domain**: TriviaQA (Joshi et al., 2017), PopQA (Mallen et al., 2023), 2WikiMultiHopQA (Ho et al., 2020), Musique (Trivedi et al., 2022), and Bamboogle (Press et al., 2023). Following **?**, we use Exact Match (EM) as the evaluation metric.

### A.8. Case Study

Table 6 presents a representative case study illustrating how BAR-RAG progressively hardens evidence composition across training iterations while operating over a fixed top-25 retrieval pool.

In Iteration 1, the selector surfaces the two [GOLDEN] documents at the top of the evidence list. One document explicitly identifies Claudio López as a retired Argentine forward who played as a main attacking player for Valencia CF, while the other confirms his role as a regular starter in Valencia's attacking line during the relevant period. The remaining documents are largely [IRRELEVANT] and do not introduce strong competing signals. As a result, the generator can answer the

question correctly with minimal reasoning effort, relying primarily on direct evidence aggregation.

In Iteration 2, the same two golden documents remain present but are no longer adjacent. They are interleaved with [MISLEADING] documents that are highly relevant at the surface level, such as profiles of Mario Kempes or summaries of Argentine forwards in La Liga. These distractors satisfy several query attributes (e.g., nationality, position, club association) but fail to jointly satisfy all constraints. Consequently, positional heuristics or shallow relevance matching become unreliable, and correct prediction requires identifying and combining the truly decisive evidence.

By Iteration 3, BAR-RAG further increases structural difficulty by pushing the two golden documents deeper into the evidence set and surrounding them with multiple misleading but plausible alternatives. The shortcut evidence remains visible but becomes insufficient: although figures such as Mario Kempes match many individual query facets, they do not simultaneously satisfy the conjunction of being retired, Argentine, a forward, and a main player for Valencia CF during the specified era. Only by consistently tracking entity identity across the dispersed golden documents can the generator arrive at the correct answer.

Across iterations, task solvability is preserved, as the same two golden documents are always present. However, the evidence structure is progressively hardened: decisive information is no longer top-ranked or contiguous, and misleading cues increasingly dominate surface relevance. This case study demonstrates that BAR-RAG improves robustness not by introducing new evidence, but by reshaping the composition and ordering of existing retrieval results to suppress shortcut reasoning and induce genuine multi-hop integration near the generator's competence boundary.

*Table 6.* Case study illustrating boundary-aware evidence selection from a fixed top-25 retrieval pool. Document IDs correspond to original retrieval ranks. Across iterations, the same two [GOLDEN] documents remain necessary to answer the question, while their relative positions are progressively dispersed and surrounded by [MISLEADING] but topically relevant noise and [IRRELEVANT] documents. This structured hardening preserves solvability while forcing multi-hop reasoning near the generator's competence boundary.

---

**Question:** Which retired Argentine footballer who played as a forward was a main player for Valencia CF?

| Iteration 1 | Iteration 2 | Iteration 3 |
|---|---|---|
| **(Doc 1)** [GOLDEN]
**Title:** Claudio López — Career Summary
**Content:** Claudio López is a retired Argentine footballer who played as a forward and was a main attacking player for Valencia CF. | **(Doc 1)** [GOLDEN]
**Title:** Claudio López — Career Summary
**Content:** Claudio López played as a main forward for Valencia CF in the late 1990s. | **(Doc 5)** [MISLEADING]
**Title:** Mario Kempes — Career Overview
**Content:** Kempes is remembered as one of the most influential Argentine forwards in Spanish football. |
| **(Doc 2)** [GOLDEN]
**Title:** Valencia CF Squad (1998–2000)
**Content:** Valencia relied on Claudio López as a regular starter in their attacking line during this period. | **(Doc 5)** [MISLEADING]
**Title:** Mario Kempes — Career Overview
**Content:** Mario Kempes is a retired Argentine forward and one of Valencia CF's most iconic historical players. | **(Doc 11)** [MISLEADING]
**Title:** Argentine Legends in La Liga
**Content:** Several Argentine forwards achieved legendary status at Spanish clubs. |
| **(Doc 3)** [IRRELEVANT]
**Title:** Valencia CF History (1990s) | **(Doc 2)** [GOLDEN]
**Title:** Valencia CF Squad (1998–2000)
**Content:** Valencia relied on a fast Argentine forward as a regular starter in their attacking system. | **(Doc 1)** [GOLDEN]
**Title:** Claudio López — Career Summary
**Content:** Claudio López is a retired Argentine forward who played for Valencia CF. |
| **(Doc 4)** [IRRELEVANT]
**Title:** Notable Footballers in La Liga | **(Doc 7)** [MISLEADING]
**Title:** Argentine Forwards in La Liga
**Content:** Several Argentine forwards, including Kempes and others, played important roles in La Liga clubs. | **(Doc 14)** [MISLEADING]
**Title:** Valencia CF Legends
**Content:** Valencia CF has featured many historically significant attacking players. |
| **(Doc 5)** [MISLEADING]
**Title:** Overview of Spanish Football Clubs | **(Doc 8)** [IRRELEVANT]
**Title:** History of La Liga Stadiums | **(Doc 2)** [GOLDEN]
**Title:** Valencia CF Squad (1998–2000)
**Content:** Valencia's main attacking options during this era included a fast Argentine forward. |

**Answer:** Claudio Javier López

---

## A.9. Prompt

This section presents the prompt templates we used for the generator and selector, in Table 7 and Table 8, respectively.

*Table 7.* Prompt template for reasoning generator.

---

**Reasoning Generator Prompt**

You are given a question and a set of retrieved documents. Your task is to answer the question **using only information from the retrieved documents**. Even for yes/no questions, you must determine the answer by reasoning from factual evidence in the documents.

**Output format (STRICT):**

- `<think>` A concise reasoning chain explaining how the answer is derived from the documents. Keep it brief (1–3 sentences). `</think>`

- `<answer>` The final answer. `</answer>`

**Evidence citation rule:**

- Whenever you use a piece of evidence from the documents in your reasoning, you **must** cite it inline as `Doc [i]`.

- You may cite one or multiple documents, but only cite documents that are actually relevant.

**Answer rules:**

- The answer should be a **short phrase** directly supported by the retrieved documents.

- Do **not** introduce external knowledge or assumptions.

- Do **not** output anything outside `<think>` and `<answer>`.

**Example (follow the style only):**
```
<think> Doc [1] states that Future Ted serves as the show's
narrator, and Doc [4] confirms that the narrator is voiced by Bob
Saget.  </think>
<answer> Ted Mosby </answer>

<Question>
<Retrieved Documents>
```

*Table 8.* Prompt template for evidence selector.

---

**Evidence Selector Prompt**

You are an expert evidence-set selector for RAG. Your goal is to select **exactly five** documents that make the question **answerable, but not trivial**. Prefer evidence sets that sit near the model's **competence boundary**: solvable with careful multi-step reasoning, yet not so direct that the answer is obvious from a single passage. You must follow the principles and output format strictly.

**Principles:**

1. **Answerability (must-have):** The selected set must contain enough information to deduce the correct answer. Do **not** select sets that make the question impossible.

2. **Non-triviality (must-have):** Avoid sets where one document directly states the answer with no integration needed. If a direct-answer passage is unavoidable for solvability, include it **only together with** supporting/context passages that require cross-document integration.

3. **Multi-hop integration:** Prefer sets that require combining at least **two** complementary clues (e.g., entity linking, temporal alignment, resolving aliases, chaining relations).

4. **Controlled noise:** Mildly conflicting or distracting details are allowed if the set remains answerable; do not include documents that are irrelevant or make the set unsolvable.

5. **Diversity:** Prefer complementary documents covering different parts of the reasoning chain, rather than near-duplicates.

**Output format (STRICT):**

- `<think>` Briefly explain which documents contain key clues, how they complement each other, and why the set is answerable but requires integration. Keep it concise (2–4 sentences). `</think>`

- `<answer>` [doc_id1], [doc_id2], [doc_id3], [doc_id4], [doc_id5] `</answer>`

**Rules:**

- Select **exactly 5** documents.

- In `<answer>`, list **only** the document identifiers in brackets, separated by commas.

- Do **not** output anything outside `<think>` and `<answer>`.

**Example (follow the style only):**
```
<think> Doc [3] provides the birthplace clue, Doc [7] gives a
timeline, and Doc [12] resolves an alias; combining them is
necessary.  Doc [5] and Doc [9] add supporting context while
introducing mild distraction, keeping the set solvable but
non-trivial.  </think>
<answer> [3], [5], [7], [9], [12] </answer>

<Question>
<Candidate Documents (Top-K)>
```

