# OpenReview forum: "Rethinking the Reranker: Boundary-Aware Evidence Selection for Robust Retrieval-Augmented Generation"
_ICML.cc/2026/Conference — ICML 2026 regular_

### Official Review · Reviewer_KUhA · 2026-03-02

**Soundness:** 2
**Presentation:** 2
**Significance:** 3
**Originality:** 3
**Overall Recommendation:** 3
**Confidence:** 4

**Summary:**

This paper addresses the brittleness of Retrieval-Augmented Generation (RAG) systems under realistic retrieval conditions. The authors propose BAR-RAG, which trains a boundary-aware selector to choose evidence sets lying in the generator’s “Goldilocks Zone”—neither too easy nor too hard—so that the generator is
challenged but can still reason correctly. A two-stage training pipeline alternates between selector training and generator fine-tuning, improving the generator’s robustness to noisy or partially relevant retrieval. Experiments show that BAR-RAG consistently improves QA accuracy and narrows the gap between retrieval recall and end-to-end performance compared to standard RAG methods.

**Compliance With Llm Reviewing Policy:**

Affirmed.

**Key Questions For Authors:**

The proposed approach improves RAG’s performance on multiple benchmark datasets; however, it remains below the state-of-the-art results achieved by methods such as RankRAG and Interact-RAG. This prompts questions regarding its practical utility for constructing a robust RAG system.

**Limitations:**

BAR-RAG introduces additional computational cost and latency due to the need for sampling multiple evidence sets and generator rollouts during selector training. This overhead may be significant when the Top-K candidate documents are large, potentially limiting real-time deployment or resource-constrained settings.

**Strengths And Weaknesses:**

**Strengths**

1.The paper is well-written and easy to understand.

2.Experiments show that BAR-RAG not only improves QA accuracy but also narrows the gap between retrieval recall and end-to-end QA performance, making it more reliable than standard RAG methods.

**Weaknesses**

1.The selector training depends on the specific generator’s ability. Changing the generator may require retraining the selector, which increases the cost of system transfer and reuse.

2.Training the Goldilocks-zone selector relies on sampling many evidence sets and generator outputs. When the Top-K candidate documents are large, this sampling can be costly and computationally expensive.

3.Missing some strong baselines (e.g., RetRobust [1], RAAT[2], InfoGain-RAG[3]).

[1]Making retrieval-augmented language models robust to irrelevant context. ICLR 2024.

[2]Enhancing noise robustness of retrieval-augmented language models with adaptive adversarial training. ACL 2024

[3]InfoGain-RAG: Boosting Retrieval-Augmented Generation through Document Information Gain-based Reranking and Filtering. EMNLP2025.
4. It would be better if the claim in Lines 73–74, that “challenging yet sufficient evidence most effectively strengthens the generator’s reasoning ability,” is supported by literature references or empirical evidence, as the current presentation provides limited justification for this motivation.

5. Typos
- It would be preferable to place Table 2 on the same page as the corresponding text description for easier reference.
- The section number on lines 766–767 failed to render correctly.

---

> ### Author Rebuttal · Authors · 2026-03-31
>
> Thank you for your valuable feedback. We address each point below:
>
> **W1**: Thank you for the helpful comment. We agree that sampling multiple evidence sets and generator rollouts during selector training introduces additional computational cost, especially with large Top-K pools. However, this overhead occurs only during training and does not affect inference latency, since the selector is not used at test time. Our approach shifts exploration cost to a one-time training procedure while preserving standard RAG efficiency. Compared with methods relying on iterative retrieval or planning, our method maintains a single-shot process without additional latency. Improving training efficiency (e.g., via better sampling or off-policy estimation) is an important future direction, and we will clarify this trade-off.
>
>
> **W2**: Thank you for this insightful comment. We agree that sampling multiple evidence sets and generator outputs during selector training introduces additional computational cost, especially with large Top-K pools. However, this cost is incurred only during training and does not affect inference efficiency, since the selector is discarded at test time. Our approach shifts exploration cost to a one-time procedure to improve robustness under noisy retrieval without increasing inference complexity. Compared with search-based alternatives that repeatedly sample and evaluate evidence per instance, our method learns a reusable selection policy that amortizes this process across training examples. Improving training efficiency (e.g., via better sampling or off-policy estimation) is an important future direction, and we will clarify this trade-off.
>
> **W3**: The methods in [1–3] are closely related and valuable. However, they are trained under different models, datasets, and retrieval settings, and do not report results under a unified setup (same backbone, retriever, and evaluation protocol). As a result, a strictly fair comparison would require substantial reimplementation and alignment, which is non-trivial within the rebuttal period. We summarize key differences:
>
> - RetRobust [1] and RAAT [2] improve robustness via filtering or adversarial training, while our method focuses on evidence selection to shape learning signals.
>
> - InfoGain-RAG [3] proposes utility-based reranking but evaluates across heterogeneous setups, making direct comparison difficult.
>
> In contrast, our work adopts a controlled setting with a shared backbone, retriever, and protocol, ensuring fair comparison. We will clarify these distinctions and include additional comparisons where feasible.
>
> **W4**: Thank you for this helpful suggestion. We agree that the motivation behind selecting “challenging yet sufficient” evidence should be better grounded. Our formulation is closely related to self-evolving reasoning and curriculum learning. For example, R-Zero trains a challenger to generate tasks near the solver’s capability boundary, while Absolute Zero shows that overly easy or unsolvable tasks provide limited learning signal, and moderate difficulty maximizes learning progress. These works suggest that learning is most effective when aligned with the model’s competence boundary. Our notion of the Goldilocks Zone is inspired by this line of work and extends it from task generation to evidence selection in RAG. We will revise the paper to make this connection more explicit.
>
> **Typos**: Thank you for pointing out these issues. We will proofread the manuscript to correct typos and adjust the layout to place Table 2 with its discussion for better readability. We will also fix the rendering issue in lines 766–767 to ensure correct section numbering.
>
> **Key Question**: Thank you for this important question. Our goal is not to directly compete with methods such as RankRAG or Interact-RAG in absolute performance, but to improve the robustness of standard RAG pipelines under realistic constraints. These methods operate under different settings: RankRAG uses substantially larger and more diverse training data, while our method follows a lightweight setup (similar to Search-R1) with ~170K examples from NQ and HotpotQA. Interact-RAG improves performance via multi-step reasoning and iterative retrieval, whereas we focus on a single-shot setting with one retrieval and one generation. Under these constraints, our method consistently improves performance without increasing inference complexity. We view this as complementary: instead of adding more data or interaction, we improve evidence quality during training, leading to more robust RAG behavior. We will clarify this positioning.
>
>
> **Limitations**: As discussed in W1, our method introduces additional cost during training (e.g., sampling and rollouts), but does not affect inference efficiency. This may increase training overhead, especially with large Top-K pools. Nevertheless, it is a one-time cost during training. Improving training efficiency will be an important direction for future work.

---

> > ### Author Rebuttal · Reviewer_KUhA · 2026-04-05
> >
> > I still believe that, given the relatively high cost of your data annotation process, the paper lacks a clear performance–cost trade-off analysis against lightweight baselines. Therefore, I am inclined to maintain my original score.

---

> > > ### Author Response · Authors · 2026-04-08
> > >
> > > Thank you for the thoughtful follow-up. We would like to clarify that our method does not rely on manual data annotation; the additional cost comes from training-time sampling and generator rollouts used to learn the selector.
> > >
> > > Importantly, this cost is incurred only once during training and is fully amortized at deployment, as the selector is not used at inference time. In contrast, many lightweight baselines (e.g., iterative retrieval or reranking-based methods) either introduce additional inference overhead or require separate training procedures, making their overall cost non-negligible as well. Existing approaches such as RetRobust, RAAT, and InfoGain-RAG also involve non-trivial costs, including additional training procedures (e.g., adversarial training, reranker optimization) or more complex inference pipelines.
> > >
> > > Our approach trades a one-time training cost for improved robustness and zero additional inference latency, which we believe is a favorable trade-off in practical RAG systems where inference efficiency is critical. We agree that a more explicit cost–performance analysis would strengthen the paper and will include additional discussion to clarify this trade-off.
> > >
> > > Moreover, our empirical gains (e.g., +10.3% EM on average) are achieved without increasing inference complexity, which distinguishes our approach from methods that rely on heavier test-time computation.

---

### Official Review · Reviewer_5qVZ · 2026-03-12

**Soundness:** 4
**Presentation:** 4
**Significance:** 4
**Originality:** 4
**Overall Recommendation:** 5
**Confidence:** 5

**Summary:**

This paper proposes BAR-RAG, a boundary-aware evidence selection framework that improves Retrieval-Augmented Generation (RAG) by selecting evidence near the generator’s competence boundary rather than purely maximizing retrieval relevance. The key insight is that traditional retrievers and rerankers often select passages that are either trivially answer-revealing or insufficient for reasoning, which leads to brittle generation and a gap between retrieval recall and answer accuracy. The method uses a two-stage training pipeline: first training a selector with RL using generator feedback to identify suitable evidence sets, and then fine-tuning the generator on the induced evidence distribution to reduce the training–inference mismatch. Experiments on multiple QA benchmarks show consistent improvements in end-to-end QA performance and robustness under noisy retrieval.

**Compliance With Llm Reviewing Policy:**

Affirmed.

**Final Justification:**

Thanks for providing additional results and clarifications, which addressed my original concerns. To reflect this, I have decided to raise my score. After reading the other reviewers’ comments and the authors’ rebuttal, I did not identify any critical issues that would change this decision.

**Key Questions For Authors:**

See Weakness above.

**Limitations:**

The authors do not have a separate "Limitations" section to discuss the limitations and potential negative societal impact of their work. I recommend that the authors add this section during the rebuttal phase.

**Strengths And Weaknesses:**

Strengths:
1. The paper introduces a new perspective on the role of rerankers in RAG systems. Instead of optimizing only retrieval relevance, BAR-RAG explicitly models the generator’s competence boundary and selects evidence that is challenging yet solvable, which provides a stronger learning signal for reasoning.
2. The proposed two-stage RL pipeline—training a selector first and then adapting the generator—effectively aligns the evidence distribution seen during training with the noisy retrieval conditions encountered during inference.
3. Experiments on several QA datasets demonstrate consistent improvements over standard RAG pipelines and reranking baselines, with notable gains on multi-hop reasoning tasks and improved robustness under retrieval noise.

Weakness:
1. Since the selector is optimized using uncertainty signals from a specific generator, it is unclear how well the learned selector generalizes to different generators. It would be interesting to see how the performance changes if the generator is replaced with a different model.
2. The notion of the Goldilocks zone is intuitive, but its boundary is not clearly defined. It would be helpful to provide a more explicit characterization of what makes evidence “too easy” or “too difficult” for the generator.
3. Since the main motivation is that evidence of different difficulty levels affects generation differently, more qualitative examples would help readers better understand what kinds of retrieved passages are preferred by the proposed selector. For example, it would be helpful to show what evidence is selected by the baseline versus BAR-RAG.

---

> ### Author Rebuttal · Authors · 2026-03-31
>
> Thank you for your valuable feedback. We address each point below:
>
> ### Table: Cross-generator transfer of selectors (EM)
>
>
> | Generator | Selector | NQ | PopQA | HotpotQA | Avg. |
> |----------|----------|----|-------|----------|------|
> | Qwen-2.5-3B (RAG) | - | 34.8 | 38.7 | 25.5 | 33.0 |
> | Qwen-2.5-3B | matched (3B) | 42.0 | 44.1 | 32.6 | 39.6 |
> | Qwen-2.5-3B | transferred (7B) | 43.5 | 45.1 | 34.2 | 40.9 |
> | Qwen-2.5-3B | transferred (LLaMA-8B) | 43.7 | 44.8 | 35.4 | 41.3 |
> | Qwen-2.5-7B (RAG) | - | 39.3 | 26.7 | 28.9 | 31.6 |
> | Qwen-2.5-7B | matched (7B) | 46.9 | 46.9 | 38.8 | 44.2 |
> | Qwen-2.5-7B | transferred (3B) | 43.1 | 44.8 | 35.7 | 41.2 |
> | Qwen-2.5-7B | transferred (LLaMA-8B) | 47.5 | 47.2 | 36.0 | 43.6 |
> | LLaMA-3.1-8B (RAG) | - | 42.7 | 28.9 | 30.3 | 34.0 |
> | LLaMA-3.1-8B | matched (LLaMA-8B) | 49.5 | 48.6 | 41.2 | 46.4 |
> | LLaMA-3.1-8B | transferred (3B) | 44.3 | 44.2 | 35.9 | 41.5 |
> | LLaMA-3.1-8B | transferred (7B) | 45.7 | 46.1 | 38.4 | 43.4 |
>
>
> **W1**: Thank you for this insightful question. Since our selector is trained from generator-dependent uncertainty signals, we agree that its transferability across generators should be examined explicitly. To this end, we conduct additional cross-generator experiments by reusing selectors trained with one model on different generators, as shown in Table. We find that matched selector-generator pairs consistently achieve the best performance, while transferred selectors still outperform the RAG baseline across datasets. This suggests that the selector indeed captures generator-specific difficulty signals, but also learns partially transferable evidence-selection patterns. We will include these results in the revised paper and clarify this trade-off more explicitly.
>
> **W2**: Thank you for this helpful suggestion. We agree that the notion of the Goldilocks Zone would benefit from a more explicit characterization. In our formulation, we operationalize evidence difficulty using the generator’s empirical correctness: for a given evidence set, we estimate the probability that the generator produces the correct answer via multiple rollouts. Evidence is considered too easy if the generator achieves near-perfect correctness (i.e., it provides little learning signal, often due to answer-revealing or shortcut-inducing context), and too difficult if the correctness is close to zero. In the RAG setting, such “too difficult” cases often arise when the retrieved documents do not contain sufficient information to answer the question, which is inevitable given that retrieval recall is imperfect. As a result, these samples provide limited learning signal and are filtered out in our framework. The Goldilocks Zone corresponds to evidence sets where the generator achieves intermediate correctness, which we use as a proxy for informative training signals. This formulation makes the notion of difficulty explicit and ties it directly to both generator behavior and retrieval quality. We will clarify this operational definition and provide more concrete examples in the revised paper.
>
> **W3**: To better illustrate the difference between standard relevance-based selection and BAR-RAG, we provide a qualitative comparison based on the example in Table 6.
> A vanilla RAG system or neural reranker would typically rank passages purely based on relevance, placing explicit answer-revealing evidence at the top. In this example, such a baseline would likely rank two passages explicitly mentioning “Claudio López” and his role at Valencia CF as the top-2 results, often accompanied by redundant or highly similar passages describing the same entity. While this evidence set is highly relevant, it makes the question trivially answerable through shallow pattern matching.
> In contrast, BAR-RAG preserves the same decisive evidence but restructures the evidence set to increase reasoning difficulty. As shown in Table 6, the golden passages are progressively dispersed and surrounded by plausible distractors (e.g., Mario Kempes, who matches many attributes such as nationality, position, and club association but does not fully satisfy the query constraints). This forces the generator to jointly reason over multiple constraints and eliminate competing candidates, rather than relying on shortcut matching.
> This example highlights a key difference: while standard reranking prioritizes relevance and tends to produce shortcut-friendly evidence configurations, BAR-RAG explicitly selects hard-but-solvable evidence that better aligns with the generator’s competence boundary. We will include this comparison in the final version.

---

> > ### Author Rebuttal · Reviewer_5qVZ · 2026-04-02
> >
> > Thanks for providing additional results and clarifications, which addressed my original concerns. To reflect this, I have decided to raise my score. After reading the other reviewers’ comments and the authors’ rebuttal, I did not identify any critical issues that would change this decision.

---

### Official Review · Reviewer_ux6W · 2026-03-13

**Soundness:** 3
**Presentation:** 2
**Significance:** 2
**Originality:** 2
**Overall Recommendation:** 4
**Confidence:** 5

**Summary:**

The authors explore improving generation quality in RAG systems by improving context selection during generator training. In fact, they train the context selector as part of their pipeline, although discard the selector for inference. The authors show relative improvements on various, mostly wikipedia-based, QA datasets.

**Compliance With Llm Reviewing Policy:**

Affirmed.

**Final Justification:**

Thank you for the follow up. I improve my score but still lean reject.

It remains important to evaluate outside of wikipedia for techniques like this where being in-domain with pretraining is a major confounder. It makes it hard to tell if the technique will generalize.

Similarly, I mentioned llama because it was the strongest option that I could tell, but I think still far behind more recent models. That being said, techniques to boost weaker models is certainly appreciated, albeit unclear if stronger or larger models will benefit. This would also benefit from having more challenging data -- maybe the llama only succeeds here because of the familiarity.

It will be great to add discussion about negatives like this to the paper. But this will be much stronger with some negative mining related baseline.

**Key Questions For Authors:**

n/a

**Strengths And Weaknesses:**

S1. This approach takes a holistic approach to context selection for training RAG generators. Previous approaches are usually simpler, focusing primarily on effective positive and negative mining. Albeit, it's unclear how the new approach compares w popular methods.

S2. Results are empirically strong, although focus primarily on wikipedia data.

S3. There is nice analysis about retrieval, including using recall to estimate upper bound answer correctness and comparing correctness with different values of K.

W1. There is very little related work mentioned. In particular, this work is very closely related to hard negative mining which is a highly studied topic area. It's plausible that simply using a negative mining technique would yield similar (or better) results than seen in the paper.

W2a. The datasets are primarily based on wikipedia, which are likely already well represented in the model's training data. Other datasets, such as CRUMB, BRIGHT, or RTEB, might be more suitable, granted this remains a hard challenge for RAG evaluation.

W2b. Similarly, there is extensive work to prevent model hallucination, a related but somewhat distinct capability compared with reasoning over retrieved documents. Using more models might show whether this recipe is similarly helpful for models that are more grounded than llama.

W3. (minor) The approach involves training the context selector (stage 1). Ablations show that w/o stage 1, quality drops from 44 to 42, which is still encouraging. Even so, it seems there are probably simple baselines to replace stage 1 that do not involve training, e.g. sampling multiple contexts and choosing the ones with most suitable reward during stage 2. This sampling could be done off-policy to aid efficiency.

---

> ### Author Rebuttal · Authors · 2026-03-31
>
> Thank you for your valuable feedback. We address each point below:
>
> **W1:** Thank you for this helpful comment. We agree that there is a conceptual connection between our approach and hard negative mining, and we will expand the related work section to better position our contribution. However, we would like to clarify that our method differs from standard negative mining in several fundamental aspects. Hard negative mining typically operates within a contrastive or retrieval training paradigm, where the goal is to identify challenging negative examples to improve representation learning. Moreover, hard negative mining implicitly assumes that increasing difficulty is always beneficial, whereas our results show that both overly easy and overly hard evidence can be suboptimal.In contrast, our approach does not focus on selecting hard negatives per se, but instead learns to shape the distribution of evidence sets based on the generator’s behavior, explicitly targeting a desired difficulty level (the Goldilocks Zone). This includes not only hard or confusing cases, but also avoiding trivially answer-revealing evidence, which is not addressed by conventional negative mining. Moreover, our method uses generator feedback via reinforcement learning to optimize a difficulty-aware objective, rather than relying on heuristic or static mining strategies. Empirically, we also compare against strong and representative baselines in both the main paper and appendix, which already cover methods that implicitly incorporate hard or challenging examples, and observe consistent improvements. We will clarify these distinctions and include a more comprehensive discussion of hard negative mining and related approaches in the revised version.
>
> **W2a and W2b:** Thank you for these helpful suggestions. We agree that Wikipedia-based datasets may partially overlap with model pretraining data. However, our goal is not to eliminate parametric knowledge, but to study how retrieval and evidence selection interact with the generator in realistic RAG settings, where such overlap is unavoidable. Importantly, we include both single-hop and multi-hop datasets (e.g., HotpotQA, MuSiQue), which require compositional reasoning beyond simple memorization. Furthermore, our counterfactual analysis in Table 4 shows that removing the selected evidence leads to a substantial performance drop, indicating that the model’s predictions are not purely driven by parametric memory, even in Wikipedia-based settings. We agree that evaluating on more challenging benchmarks (e.g., CRUMB, BRIGHT, RTEB) would be valuable, and we will include this discussion as future work.
>
> Regarding model diversity, our primary goal is to isolate the effect of evidence selection and training dynamics, rather than differences across model architectures. To this end, we already conduct experiments across multiple widely used open models, including Qwen-2.5-3B-Instruct, Qwen-2.5-7B-Instruct, and LLaMA-3.1-8B-Instruct, covering different model families and scales. We observe consistent improvements across all these models, suggesting that the proposed training paradigm is not tied to a specific architecture or model configuration. While we agree that it would be interesting to further evaluate models with stronger grounding or reduced hallucination, our current results already demonstrate that the effectiveness of our approach generalizes across diverse base models. We will clarify this point in the revised paper and include a more explicit discussion on model generalization.
>
> **W3:** Thank you for this insightful suggestion. We agree that simpler alternatives, such as sampling multiple evidence sets and selecting those with desirable rewards, could be considered as a heuristic baseline. However, we would like to emphasize that such approaches differ fundamentally from our method. Our stage 1 explicitly learns a policy to shape the evidence distribution, enabling generalization across queries, whereas sampling-based approaches operate as per-instance search heuristics without learning a reusable selection strategy. In particular, off-policy sampling relies on the quality and coverage of the candidate pool, and may become inefficient or unstable when the search space is large, as it does not adaptively improve over time. In contrast, our RL-based selector amortizes this search process by learning to directly generate evidence sets near the desired difficulty boundary, leading to more consistent behavior. We also note that our ablation (w/o stage 1) still shows reasonable performance, which suggests that stage 1 is not strictly required, but consistently improves results by better aligning the evidence distribution with the training objective. Exploring lightweight or hybrid alternatives (e.g., combining sampling with learned policies) is an interesting direction for future work, and we will include this discussion in the revised paper.

---

> > ### Author Rebuttal · Reviewer_ux6W · 2026-04-04
> >
> > Thank you for the follow up. I improve my score but still lean reject.
> >
> > It remains important to evaluate outside of wikipedia for techniques like this where being in-domain with pretraining is a major confounder. It makes it hard to tell if the technique will generalize.
> >
> > Similarly, I mentioned llama because it was the strongest option that I could tell, but I think still far behind more recent models. That being said, techniques to boost weaker models is certainly appreciated, albeit unclear if stronger or larger models will benefit. This would also benefit from having more challenging data -- maybe the llama only succeeds here because of the familiarity.
> >
> > It will be great to add discussion about negatives like this to the paper. But this will be much stronger with some negative mining related baseline.

---

### Official Review · Reviewer_HH5m · 2026-03-13

**Soundness:** 2
**Presentation:** 3
**Significance:** 3
**Originality:** 3
**Overall Recommendation:** 5
**Confidence:** 4

**Summary:**

The authors tackle an important problem, going beyond semantic relevance matching for selecting documents in RAG pipelines. Rather, authors incorporate generator outcomes and reframe the reranker as a selector that targets to prioritize documents that are close to the boundary of the goldilocks zone with respect to the generator, that is evidences that lead to solvability of the task while at the same time ensuring they do not lead to trivial reasoning in the generator. Experiments on QA datasets that require complex reasoning (like 2Wiki) authors demonstrate gains in QA performance.

**Compliance With Llm Reviewing Policy:**

Affirmed.

**Final Justification:**

As most of my concerns are addressed I am happy to increase my score to 5.

**Key Questions For Authors:**

Please refer to my questions in the weaknesses section.

**Limitations:**

yes

**Strengths And Weaknesses:**

*Strengths:*

1) Reformulating the reranker to select evidence close to the boundary of solvability is a pretty interesting idea and also a different mechanism of employing generator uncertainty than other works in generator uncertainty / feedback-based ranking approaches, I am aware of.

2) The motivation  not just relying on semantic relevance is important as the core focus should be ranking evidence that is useful for the downstream generator. In that sense, the paper tackles an important problem that will be significant to the IR and NLP community.

3) The reward design in 2.4 is intuitive and covers the different key aspects of RAG-based tasks. However, I believe faithfulness of attributions has to be accounted for as detailed in my comment in weaknesses section.

The results for larger generators with BAR-RAG gives quite impressive gains in the final QA task. The ablations also demonstrate the need for proposed components.

*Weaknesses:*

1) The failure mode stated in Introduction :``Current retrievers optimize exclusively for query-document relevance (Wang et al., 2022; Zhang et al., 2025), creating two systematic failure modes: they prefer trivial, answer-revealing passages that encourage shortcut learning, ” is also an artifact of the dataset no? For instance, it is well documented [1] that HotpotQA really does not require multi-hop reasoning for a significant number of questions, and the reasoners can just adopt a shortcut, which was an artifact of how the dataset was constructed and the lack of checks mandating multi-hop reasoning. In fact, this is the motivation stated in the formation of MusiqueQA, where authors explicitly state this and perform rigorous data validation to avoid such scenarios. While i agree with the motivation that the re-ranker should not just prioritize evidence based on semantic relatedness to query, but rather based on the evidence’s usefulness for downstream reasoning by generator, the above motivation of shortcuts is a bit misleading without concrete example and precise description of whether the shortcuts are an artifact of the data or the pipeline. I would recommend that authors elaborate on this further for clarity or focus on the evidence usefulness aspect with regard to reasoning as motivation.

2) There are some more relevant baselines with regard to “with retrieval” setting, like Self-Ask [2], Searchain [3], which also incorporate iterative retrieval with feedback from the generator in the form of follow-up questions / plans, which I believe are relevant baselines to the current work. I would like to see some discussion or comparison to the mentioned works. Additionally, several recent methods focus on uncertainty quantification in the RAG setup to gauge the usefulness of retrieved or ranked documents for downstream reasoning and consequently the answering performance [4,5,7]. I believe a discussion of these works are also relevant since the proposed approach also deals with generator uncertainty.

3) I believe relating the definition of choosing evidence that leads to near target level correctness ( the boundary) to solvability, yet non-trivial reasoning is not necessarily strictly true always. For instance, in some cases, the generator could generate the right answer from parametric memory rather than evidence. While you do employ a citation reward to train the generator, this reward focuses more on number of attributions than checking if they are faithful. There are prior works demonstrating that citations are not necessarily faithful in RAG systems [6]. Even if citations may seem right, they could be post-rationalized by the generator while it actually generates the answer from parametric memory, and your evidence may not be able to correctly satisfy the criteria for solvability and could be incorrectly ranked evidence. I think faithfulness based reward and not just answer correctness is also required to ensure that evidence chosen near this generator reward boundary satisfy criteria for solvability and non-triviality (the goldilocks zone).


[1] MuSiQue: Multihop Questions via Single-hop Question Composition
Harsh Trivedi Niranjan Balasubramanian Tushar Khot† Ashish Sabharwal†

[2] Measuring and Narrowing the Compositionality Gap in Language Models
Ofir Press, Muru Zhang, Sewon Min, Ludwig Schmidt, Noah Smith, Mike Lewis

[3] Search-in-the-Chain: Interactively Enhancing Large Language Models with Search for Knowledge-intensive Tasks
Shicheng Xu, Liang Pang, Huawei Shen, Xueqi Cheng, Tat-Seng Chua

[4] Uncertainty Quantification in Retrieval Augmented Question Answering
Laura Perez-Beltrachini, Mirella Lapata

[5] Why Uncertainty Estimation Methods Fall Short in RAG: An Axiomatic Analysis
Heydar Soudani, Evangelos Kanoulas, Faegheh Hasibi

[6] Correctness is not Faithfulness in Retrieval Augmented Generation Attributions
Authors: Jonas Wallat, Maria Heuss, Maarten de Rijke, Avishek Anand

[7] Adaptive Retrieval without Self-Knowledge? Bringing Uncertainty Back Home

---

> ### Author Rebuttal · Authors · 2026-03-31
>
> Thank you for your valuable feedback. We address each point below:
>
> **W1:** Thank you for this insightful comment. We agree that certain datasets (e.g., HotpotQA) contain shortcut artifacts due to their construction, and we will revise the introduction to avoid over-attributing this phenomenon solely to retrievers or rerankers. That said, our motivation is not limited to dataset-specific biases. Rather, we aim to highlight a more general issue: existing retrieval and reranking methods optimize for relevance, but do not consider whether the selected evidence is actually useful for downstream reasoning by the generator. In practice, even highly relevant evidence can be either trivially answer-revealing (encouraging shortcut behavior) or insufficient for multi-step inference, and this issue persists beyond dataset artifacts. For example, even in more carefully curated datasets designed to reduce shortcuts (e.g., MuSiQue), we still observe a gap between retrieval recall and QA accuracy (Figure 2), suggesting that relevance alone is not sufficient for effective reasoning. We also note that this limitation has been increasingly recognized in recent work such as Search-R1, Search-O1, and DynamicRAG, which implicitly demonstrate that evidence usefulness for reasoning is more critical than semantic relatedness alone. Our work builds on this insight but differs in that we explicitly model and optimize the evidence difficulty distribution via reinforcement learning, targeting evidence near the generator’s competence boundary. Following the reviewer’s suggestion, we will revise the introduction to clarify the distinction between dataset artifacts and pipeline-level limitations, and focus the motivation more clearly on evidence usefulness for reasoning.
>
> **W2:** Thank you for the helpful suggestions. We agree these works are relevant and will expand the discussion. Methods such as Self-Ask [2] and Search-in-the-Chain [3] focus on test-time iterative retrieval, while our approach operates at training time, learning a selector to shape the evidence distribution and improve robustness to noisy retrieval without requiring iterative inference. Thus, these methods are complementary rather than directly comparable.
> For uncertainty-based approaches [4,5,7], prior work typically uses uncertainty as a post-hoc signal for filtering or calibration, whereas we use generator uncertainty as a training objective to explicitly optimize evidence difficulty around the competence boundary.
> We also note that prior work highlights a gap between correctness and faithfulness in RAG [6], indicating that relevance or correctness alone does not capture evidence usefulness for reasoning. Finally, we already compare against strong baselines in the main results and appendix; our goal is to improve standard RAG pipelines without increasing inference cost. We will clarify these distinctions and expand related work in the revision.
>
> **W3:** Thank you for the thoughtful comment. We agree that the connection between the Goldilocks Zone (i.e., targeting a specific correctness level) and the notions of “solvability” and “non-trivial reasoning” is not strictly guaranteed in all cases. In particular, as pointed out by prior work [6], the generator may sometimes produce correct answers from parametric memory and post-rationalize citations, which can weaken the faithfulness of evidence usage. Our formulation does not assume that correctness perfectly implies faithful grounding; instead, we use empirical correctness as an operational proxy for difficulty, capturing how challenging an evidence set is for the generator in practice. Importantly, we complement this with additional analysis to verify that the model’s predictions are indeed dependent on the selected evidence. As shown in Table 4, removing the originally cited documents leads to a substantial drop in performance across datasets, while keeping only these cited documents largely preserves performance. This counterfactual evidence-dependence analysis suggests that the model’s predictions rely meaningfully on the selected evidence rather than being purely driven by parametric memory or post-hoc rationalization. Moreover, even in the presence of potential parametric shortcuts, we empirically observe that evidence sets near the target correctness boundary induce more informative and balanced generator behavior, avoiding both trivial answer-revealing contexts and consistently unsolvable ones. We acknowledge that our current citation reward does not explicitly enforce faithfulness, and incorporating stronger faithfulness-aware signals is an interesting direction for future work. We will clarify this distinction in the revised paper and moderate the claim accordingly to emphasize that our method models difficulty with respect to generator behavior, rather than strictly guaranteeing faithful evidence grounding.

---

### Decision · Program_Chairs · 2026-04-30

**Decision:**

Accept (regular)

**Comment:**

- This paper proposes BAR‑RAG, a boundary‑aware evidence selection framework that improves RAG robustness by selecting evidence useful for downstream reasoning rather than purely maximizing retrieval relevance. Reviewers found this to be a novel and meaningful direction.

- During rebuttal, the authors clarified motivation, expanded discussion of related work, and provided additional transfer experiments, addressing several reviewer concerns and leading to score increases. However, limitations remain, including generator‑dependent selector training, additional training‑time cost, and evaluation primarily on Wikipedia‑based datasets, which leaves generalization and cost–performance tradeoffs less clear.

- Overall, the approach is technically sound and improves robustness without increasing inference complexity, but would benefit from broader evaluation. I therefore recommend **Weak Accept**.